# Comparative Analysis of the Gut Microbiota of Bat Species with Different Feeding Habits

**DOI:** 10.3390/biology13060363

**Published:** 2024-05-22

**Authors:** Diego Antonio Mena Canata, Mara Silveira Benfato, Francielly Dias Pereira, María João Ramos Pereira, Fernanda Schäfer Hackenhaar, Michele Bertoni Mann, Ana Paula Guedes Frazzon, Pabulo Henrique Rampelotto

**Affiliations:** 1Biophysics Department, Universidade Federal do Rio Grande do Sul, Porto Alegre 91501-970, Brazil; 2Graduate Program in Cellular and Molecular Biology, Universidade Federal do Rio Grande do Sul, Porto Alegre 91501-970, Brazil; 3Graduate Program in Animal Biology, Laboratory of Evolution, Systematics and Ecology of Birds and Mammals, Universidade Federal do Rio Grande do Sul, Porto Alegre 91501-970, Brazil; 4Department of Medical Biosciences, Umeå University, SE-90185 Umeå, Sweden; 5Graduate Program in Agricole and Environmental Microbiology, Universidade Federal do Rio Grande do Sul, Porto Alegre 91501-970, Brazil; 6Bioinformatics and Biostatistics Core Facility, Instituto de Ciências Básicas da Saúde, Universidade Federal do Rio Grande do Sul, Porto Alegre 91501-970, Brazil

**Keywords:** microbiome, nectarivore, frugivore, insectivore, vampire bat, diet

## Abstract

**Simple Summary:**

Bats represent a diverse and ecologically significant mammalian group characterized by different feeding habits. Despite the gut microbiota influence on feeding habits, microbiome studies are scarce in South American bats, in a region considered a hot spot for bat biodiversity. Our study in Southern Brazil compared the gut microbiota of four bat species with different diets: nectarivorous, frugivorous, insectivorous, and hematophagous. We found that each species had unique gut microbiotas linked to their dietary habits, impacting their metabolic potentials. The presence of potentially harmful bacteria varied with feeding habits, suggesting a correlation between diet and microbial pathogens in bats. These insights emphasize the importance of preserving diverse habitats and food sources to support the conservation of bats and their ecosystems.

**Abstract:**

Bats are a diverse and ecologically important group of mammals that exhibit remarkable diversity in their feeding habits. These diverse feeding habits are thought to be reflected in the composition and function of their gut microbiota, which plays important roles in nutrient acquisition, immune function, and overall health. Despite the rich biodiversity of bat species in South America, there is a lack of microbiome studies focusing on bats from this region. Such studies could offer major insights into conservation efforts and the preservation of biodiversity in South America. In this work, we aimed to compare the gut microbiota of four bat species with different feeding habits from Southern Brazil, including nectarivorous, frugivorous, insectivorous, and hematophagous bats. Our findings demonstrate that feeding habits can have a significant impact on the diversity and composition of bat gut microbiotas, with each species exhibiting unique metabolic potentials related to their dietary niches. In addition, the identification of potentially pathogenic bacteria suggests that the carriage of microbial pathogens by bats may vary, depending on feeding habits and host-specific factors. These findings provide novel insights into the relationship between bat feeding habits and gut microbiota composition, highlighting the need to promote diverse habitats and food sources to support these ecologically important species.

## 1. Introduction

Bats are a diverse group of mammals that perform vital functions in maintaining ecosystem balance, which encompasses the tasks of pollinating plants, dispersing seeds, and controlling insect populations [1]. Despite their ecological importance, bats are currently facing numerous threats, including emerging infectious diseases, climate change, and habitat loss [2]. One factor that may impact the health and survival of bat populations is their feeding habits.

Bats have evolved to consume a wide variety of foods, including nectar, fruit, insects, and even blood. These diverse feeding habits have been linked to differences in bat morphology, physiology, and behavior [3]. For example, nectarivorous bats have long, slender snouts and tongues adapted for drinking nectar from flowers, while insectivorous bats have sharp teeth and echolocation abilities that enable them to catch and consume fast-moving prey [4]. Hematophagous bats have specialized adaptations for feeding on blood, including heat sensors on their noses to locate warm-blooded prey and anticoagulants in their saliva to facilitate feeding [5].

In addition to these morphological and physiological adaptations, recent research has suggested that the gut microbiota of bats may also play a role in their feeding habits [6]. The gut microbiota comprises a complex assemblage of microorganisms residing in the digestive tracts of animals, including bats. These microorganisms have important functions, such as aiding in digestion, modulating immune function, and preventing the colonization of harmful pathogens [7].

Different studies have suggested that the gut microbiota of bats can vary based on their feeding habits, with significant changes observed in the composition and diversity of bacterial populations [8,9,10]. These investigations have demonstrated that the gut microbiota of bats undergo significant variations in bacterial diversity and composition in response to their specific feeding habits. For instance, insectivorous bats harbor microbiota enriched in bacterial taxa capable of digesting chitin from insect exoskeletons, while frugivorous bats possess microbial communities specialized in metabolizing fruit-derived compounds [11]. This specialization reflects adaptations to the digestion of distinct dietary components corresponding to their feeding habits. Furthermore, environmental factors, geographic locations, and seasonal dynamics have been shown to exert additional influences on the gut microbiota of bats [12,13]. These factors contribute to the variability observed in the microbial communities of bats across different habitats and periods, highlighting the dynamic nature of bat gut microbiota and emphasizing the multifaceted factors shaping their microbial ecology [14,15,16]. Understanding these dynamics not only enhances our comprehension of bat ecology but also sheds light on potential implications for bat health, including disease susceptibility and conservation efforts.

South America is home to a wide variety of bat species with different feeding habits [17]. Understanding the gut microbiota of these bats can provide valuable insights into the adaptations of these mammals to their specific dietary niches, as well as shed light on the ecological roles that bats play in their respective ecosystems. However, despite the rich biodiversity of bat species in South America, with Brazil in particular hosting approximately 15% of the world’s bat diversity, the literature lacks microbiome studies of bats from this region [18,19,20]. The lack of comprehensive studies on the gut microbiota of bat species from South America represents a significant gap in our understanding of the microbial ecology of bats. In this study, we aimed to compare the bacterial communities in the guts of four bat species from Southern Brazil with different feeding habits, including nectarivorous, frugivorous, insectivorous, and hematophagous bats. Our findings provide novel insights into the relationship between bat feeding habits and gut microbiota composition, with significant implications for understanding the ecology and conservation of bat communities.

## 2. Materials and Methods

### 2.1. License Authorization and Ethical Approval

The bats included in this work were captured with permission granted by CONCEA (the National Council for the Control of Animal Experimentation, No. 33339) and SISBIO (the Brazilian Biodiversity Information and Authorization System, No. 47202-1). Ethical approval was provided by CEUA/UFRGS (the University’s Ethics Committee on the Use of Animals, No. 28645). 

### 2.2. Animals and Samples Collection

Figure 1 shows the locations of the collection sites and coordinates. Our previous study provided a detailed description of the capture procedure [21]. To summarize, over the summer of 2018 and the winter of 2019, 33 adult male bats were caught in southern Brazil. The captured bat species were *Glossophaga soricina* (*n* = 7, nectarivore, autumn 2019), *Sturnira lilium* (*n* = 10, frugivore, winter 2019), *Molossus molossus* (*n* = 10, insectivore, summer 2018), and *Desmodus rotundus* (*n* = 6, hematophagous, summer 2018). The degree of ossification in wing elements was used to differentiate adult bats [22]. To make sure the bats were fasted, they were caught early in the night, and they were euthanized on-site with an intraperitoneal injection of xylazine (10 mg/kg) and ketamine (60 mg/kg). The bats were then immediately placed into plastic bags, frozen using liquid nitrogen, and then kept on dry ice until they could be transported to the nearby facility. Upon arrival at the facility, they were stored in a freezer at −80 °C for preservation.

### 2.3. DNA Extraction and 16S rRNA Amplicon Sequencing 

In the lab, the intestinal samples were thawed to extract the fecal content. Homogenization of the intestinal contents was performed, and 200 mg of each sample was utilized for genomic DNA extraction using the E.Z.N.A. Stool DNA Kit (Omega Bio-Tek, Norcross, GA, USA), following the manufacturer’s instructions. The concentration of DNA was determined using a Qubit 3.0 fluorometer (Thermo Fisher Scientific, Waltham, MA, USA).

To target the V4 hypervariable region of bacterial 16S rRNA, the 515F/806R primer pair (515F: 5’-GTGCCAGCMGCCGCGGTAA-3’ and 806R: 5’-GGACTACHVGGGTWTCTAAT-3’) was employed. The PCR reaction was conducted in a 50 µL total volume containing 1× buffer, 0.2 mM dNTPs, 0.2 µM of each primer, 1.5 mM MgCl_2_, 2U Platinum Taq DNA polymerase, 12.5 ng genomic DNA, and water to complete the volume. The PCR amplification was carried out using the Biorad MyCycler Thermal Cycle under the following conditions: an initial denaturation at 94 °C for 3 min, followed by 30 cycles of denaturation at 94 °C for 30 s, annealing at 55 °C for 30 s, extension at 72 °C for 30 s, and a final extension at 72 °C for 5 min. Purification of the PCR products was performed using Agencourt AMPure XP (Beckman Coulter, Indianapolis, IN, USA) according to the manufacturer’s protocol. Indexes were added to the DNA libraries following the instructions provided by Illumina. The sequencing process was carried out using the MiSeq Reagent Kit v2 (Illumina, San Diego, CA, USA).

### 2.4. Bioinformatics Analyses

The sequence data obtained from the Miseq System underwent processing using a customized pipeline in Mothur v.1.48.0 [23]. Initially, the sequences were stripped of barcodes and primers (with no allowed mismatches), followed by the application of a quality filter to remove low-quality reads. The quality control involved trimming reads with low quality (Q < 30), incorrect length (minlength = 270 pb, maxlength = 300 pb), ambiguous bases (maxambig = 0), or homopolymers longer than 6 bp. VSEARCH v. 2.26.0 was used to identify and eliminate potentially chimeric sequences [24]. Additionally, singletons were excluded to prevent the inclusion of sequences that could be spurious due to PCR or sequencing errors [25].

After quality filtering, the remaining sequences were then clustered into amplicon sequencing variants (ASVs) and classified against the SILVA v.138 reference database [26]. Sequences that could not be identified (referred to as “unknown” sequences), as well as sequences assigned to chloroplasts, mitochondria, and eukaryotes, were excluded from further analysis. Subsequently, the resulting ASV table was adjusted to the size of the smallest library. The sequence dataset was further analyzed using R v.4.0.0 (The R Foundation for Statistical Computing, Vienna, Austria), making use of phyloseq and MicrobiomeAnalystR packages.

### 2.5. Microbial Community and Statistical Analysis 

Alpha diversity measures, including the number of observed taxa, ACE, and Shannon index, were used to assess the diversity within the microbial communities. The statistical significance of these diversity indices was evaluated using the multivariate Kruskal–Wallis test. For the comparison of overall dissimilarities among bacterial communities (beta diversity), principal coordinates analysis (PCoA) was conducted. A dissimilarity matrix based on the Bray–Curtis metric was computed for each pair of samples. The analysis of similarities (ANOSIM) multivariate test was employed to determine the statistical significance of the observed sample grouping in the PCoA results. Microbial composition was expressed as the relative abundance. Furthermore, to explore further distinctions among the microbial communities, a clustering technique based on the Bray–Curtis dissimilarity was employed, and the outcome was visualized using a dendrogram. A Venn dendrogram was generated using InteractiVenn [27].

The linear discriminant effect size (LEfSe) method was used to identify differentially abundant microbial taxa at the genus level [28]. Microbial taxa with a logarithmic LDA score exceeding ±2.0 and a corrected *p*-value below 0.05 were significantly differentially abundant. 

### 2.6. Metabolic Prediction

Predictions of metabolic pathways relied on ASV sequences and their prevalence in PICRUSt2 [29]. PICRUSt2’s functional annotations stemmed from the MetaCyc database. LEfSe identified pathways with differential abundance. Significance was attributed to variations meeting a logarithmic LDA score threshold of ±4.0 and a corrected *p*-value of less than 0.05.

## 3. Results

### 3.1. Microbial Classification

The ASV table consisted of 3,398,991 high-quality sequences obtained from 33 samples. On average, each sample contained approximately 102,999 sequences (Appendix A). Good’s coverage, a measure of sequence coverage, was found to be 99.9% ± 0.05 for all samples, indicating a high level of coverage (Appendix A). The rarefaction curves demonstrated that as the number of sequences increased, the richness and diversity of ASVs reached a stable state in an unbiased manner for each sample (Appendix A). In total, 646 ASVs were identified and categorized into 15 phyla, 127 families, and 231 genera (Appendix A).

### 3.2. Microbial Diversity

No significant differences were observed among the four bat species in the number of microbial taxa (*p*-value = 0.46) and ACE (*p*-value = 0.24). However, significant differences were observed for Shannon (*p*-value = 0.001) and Simpson (*p*-value = 0.001), with nectarivorous and insectivorous bats presenting a higher diversity than frugivorous and hematophagous bats (Figure 2). 

Beta diversity analysis indicated that gut bacterial communities significantly differed among the four bat species (Figure 3), which was confirmed by ANOSIM (r = 0.93, *p*-value = 0.001).

### 3.3. Microbial Composition

*Firmicutes* (72%), *Proteobacteria* (14%), *Campylobacterota* (5%), *Actinobacteriota* (3%), and *Verrucomicrobiota* (2%) were the most abundant phyla in all samples (Appendix A). The relative abundance of other phyla did not exceed 1% in all four bat species. The dominant families with a relative abundance greater than 10% were *Mycoplasmataceae* (15%) *Acholeplasmataceae* (12%), *Streptococcaceae* (11%), *Clostridiaceae* (10%), and *Peptostreptococcaceae* (10%) (Appendix A). At the genus level, *Mycoplasma* (0.13%), *Acholeplasmataceae*_unclassified (12%), *Streptococcus* (10%), *Peptostreptococcaceae*_unclassified (9%), and *Enterobacteriaceae*_unclassified (5%) were the genera with the highest relative abundance (Appendix A).

The analysis using LEfSe demonstrated significant variations in the gut microbiota among the four bat species at all taxonomic levels. A total of 22 genera exhibited differential abundance among these bat species (Figure 4); seven in nectarivorous bats (*Clostridium*_ss_1, *Enterobacteriaceae*_unclassified, *Ureaplasma*, *Neisseriaceae*_unclassified, *Clostridiaceae*_unclassified, *Actinomyces*, and *Terrisporobacter*), three in frugivorous bats (*Acholeplasmataceae*_unclassified, *Campylobacter*, and *Helicobacter*), nine in insectivorous bats (*Erysipelotrichales*_unclassified, *Actinobacteria*_unclassified, *Fusobacterium*, *Atopobium*, *Lactobacillales*_unclassified, *Pasteurellaceae*_unclassified, *Bacilli*_unclassified, Candidatus_Arthromitus, and *Ligilactobacillus*—differentially abundant in insectivorous bats, though highly abundant in one sample from hematophagous bats), and two in hematophagous bats (*Peptostreptococcaceae* and *Edwardsiella*). *Chlamydia* was only highly abundant in one sample from insectivorous bats.

### 3.4. Shared and Unique Microbiota

Of the 231 genera identified in this study, only 45 were shared by all four bat species (Figure 5) (Appendix A). Insectivorous bats presented the highest number of unique genera (42), followed by hematophagous (22), frugivorous (11), and nectarivorous (11) bats (Appendix A). 

### 3.5. Potential Bacterial Pathogens

To study the presence of opportunistic pathogens in bat samples, ASVs from known opportunistic pathogenic genera were identified, and the results are summarized in Figure 6. In total, 78 ASVs from 12 potentially pathogenic genera were detected with different proportions, including *Bartonella*, *Brucella*, *Campylobacter*, *Chlamydia*, Clostridium_ss_1, *Mycobacterium*, *Mycoplasma*, *Peptostreptococcaceae*_unclassified, *Pseudomonas*, *Staphylococcus*, *Streptococcus*, and *Treponema*. *Bartonella* was only observed in nectarivorous bats, while *Brucella* was sporadically observed in three bat species. While ASV0003 and ASV0068 from *Campylobacter* were observed in nearly all samples from frugivorous and insectivorous bats, the other five ASVs from this genus were consistently observed in four frugivorous samples (F2, F3, F5, and F6). While ASV0011 from *Chlamydia* was consistently detected in frugivorous and insectivorous bats, the other two ASVs from this genus were observed in only one sample. While ASV0035 from *Clostridium* sensu stricto (cluster I) was detected in all four bat species, the other ASVs from this genus were consistently observed in three nectarivorous samples (N1, N2, and N7). Among the 22 ASVs from *Mycoplasma*, only 6 were consistently detected in 1 of the bat species, while the other 16 were sporadically observed in 1–3 samples. While ASV0058 from *Pseudomonas* and ASV0017 from *Staphylococcus* were consistently detected in all four bat species, the other ASVs from these genera were sporadically observed. Several ASVs from *Streptococcus* were consistently detected in all four bat species, and the four ASVs from *Treponema* were sporadically observed only in insectivorous and hematophagous bats. Other potential pathogens already reported in bat studies (e.g., *Bordetella*, *Enterococcus*, *Escherichia*, *Salmonella*, *Shigella*, *Yersinia*, and *Vibrio*) were not observed in any samples.

### 3.6. Functional Profile

In total, 21 predicted metabolic pathways exhibited differential abundance among the four bat species (Figure 7). Deoxyribonucleotides biosynthesis was higher in hematophagous bats and lower in insectivorous bats. Adenosine ribonucleotides biosynthesis and UMP biosynthesis were higher in frugivorous and hematophagous bats. Nucleotide degradation was higher in hematophagous and insectivorous bats and lower in frugivorous bats. Pathways related to Vitamin B12 metabolism were higher in hematophagous bats and lower in frugivorous bats. Regarding carbohydrate metabolism, sucrose degradation and pentose phosphate pathways were higher in frugivorous bats and lower in hematophagous bats. Myo-, chiro-, and scillo-inositol degradations were higher in nectarivorous bats and lower in frugivorous bats. Urate biosynthesis was higher in hematophagous and insectivorous bats and lower in frugivorous bats. Biotin biosynthesis was excessively higher in hematophagous bats, compared to other bats. Methanogenesis from acetate was also excessively higher in hematophagous bats.

## 4. Discussion

The results of this study demonstrate that the feeding habits of bats can have a significant impact on the diversity of their communities. While no difference was observed in the number of observed microbial taxa and ACE, nectarivorous and insectivorous bats were found to have higher diversity than frugivorous and hematophagous bats when measured by the Shannon index. These results indicate that the different feeding habits do not affect the community richness (observed taxa and ACE), but they significantly influence the community diversity of these bats (Shannon and Simpson). 

The bat species known as *G. soricina* is distributed across Latin America, has a diet that includes nectar from flowers and floral parts, and inhabits various ecosystems, such as tropical forests, savannas, and urban areas [30]. *S. lilium* is found in various habitats across Central and South America and mainly feeds on fruits from the Solanaceae family, indicating a preference for certain types of fruit [31]. Widely distributed across the Americas, *M. molossus* occupies diverse habitats, such as forests, grasslands, and urban areas. This bat species is an insectivore that consumes various insects, with a preference for Coleoptera (beetles) [32]. The common vampire bat *D. rotundus*, also present in Latin America, exclusively feeds on the blood of domestic cattle [33].

The lack of significant differences in observed microbial taxa and ACE among the four bat species indicates that the species richness of bat communities may be relatively stable across different feeding habits, which is likely because bats have evolved to utilize a diverse array of food sources [34]. This suggests that these bat species, despite having different diets, have managed to adapt to their respective habitats and ecological niches to effectively find and exploit their food sources, allowing them to coexist without directly competing for limited resources. This ability to utilize a variety of food sources may have contributed to the stability of species richness in the studied bat communities.

However, the significant differences observed in the Shannon and Simpson indices indicate that the diversity of bat communities can be influenced by feeding habits. The higher diversity observed in nectarivorous and insectivorous bats (compared to frugivorous and hematophagous) suggests that these feeding habits may be associated with different ecological roles and preferences for certain habitats or resources. For example, nectarivorous bats may be more dependent on flowering plants and their associated microhabitats, while insectivorous bats may be more closely linked to the availability of prey in certain areas [35]. On the other hand, the low diversity observed in hematophagous bats indicates a specialized feeding habitat in consuming a single food resource, e.g., blood [36]. Despite their broad diet of fruits [37], low bacterial diversity was observed in frugivorous bats. One possible explanation for this observation is that the bat’s digestive system may have evolved to efficiently digest and extract nutrients from its diet, resulting in a reduction in bacterial diversity in the gut. This phenomenon has been observed in other animals, where the host’s digestive system selectively favors certain bacterial species over others, resulting in a low diversity of bacterial communities [38]. 

These results have important implications for the conservation of bat communities, as different feeding habits may require different management strategies. For example, protecting and promoting the availability of flowering plants may be important for the conservation of nectarivorous bats, while efforts to maintain insect populations may be critical for the survival of insectivorous bats. Furthermore, the results suggest that the loss of certain feeding habits could have significant impacts on the diversity of bat communities, underscoring the importance of maintaining diverse habitats and food sources to support bat populations [39].

The identification of specific genera that are differentially abundant among the bat species may provide insights into the functional roles of the gut microbiota in bat physiology and ecology. Our findings are consistent with previous studies that have shown that the gut microbiota of nectarivorous bats is enriched in microbial taxa involved in carbohydrate metabolism, such as *Clostridium* and *Actinomyces*, whereas the gut microbiota of frugivorous bats is enriched in microbial taxa associated with fruit fermentation, such as *Campylobacter* and *Helicobacter* [40,41]. Insectivorous bats, on the other hand, have gut microbiota dominated by microbial taxa that are involved in protein and fat digestion, such as *Bacilli* and *Erysipelotrichales* [42]. The presence of Peptostreptococcaceae and *Edwardsiella* in hematophagous bats may be related to their blood-feeding behavior. *Peptostreptococcaceae* is a group of anaerobic bacteria commonly found in the gut microbiota of animals and is associated with the degradation of complex organic matter, including blood [43]. *Edwardsiella*, on the other hand, is a genus of Gram-negative bacteria that have been found in the guts of various animals, including domestic animals [44]. The presence of *Edwardsiella* in the guts of hematophagous bats may be related to their ability to feed on blood, which could potentially expose them to a variety of niche-restricted microorganisms. 

Considering the high number of bacterial pathogens that may be carried by bats [45], we further analyzed in detail some taxa from our dataset to search for potential bacteria pathogens. The 78 ASVs from 12 potentially pathogenic genera indicate that the number of ASVs from each potentially pathogenic genus, as well as their distribution in samples, is highly variable. The observation that *Bartonella* was only detected in nectarivorous bats and *Brucella* was sporadically observed in three bat species suggests that different feeding habits may influence the carriage of specific bacterial pathogens. Similarly, the observation that certain ASVs from *Campylobacter* and *Clostridium* sensu stricto were consistently observed in certain bat species indicates that host-specific factors may play a role in the bacterial carriage. Worthy of note, while several potentially pathogenic genera were detected in this study, other commonly reported genera in bat studies (e.g., *Bordetella*, *Enterococcus*, *Escherichia*, *Salmonella*, *Shigella*, *Yersinia*, and *Vibrio* [46]) were not observed in any sample. This suggests that the carriage of potentially pathogenic bacteria by bats may vary depending on geographical location, bat species, and other environmental factors. In addition, the different profiles of ASVs from the same genus suggest the presence of different species, which increases the diversity of potentially pathogenic microbial taxa. The identification of potentially pathogenic microbial taxa highlights the importance of studying the gut microbiota of bats from a public health perspective [47]. Some of these microbial taxa may pose a risk to both bat and human health, and understanding their ecology and potential transmission pathways is important for disease surveillance and prevention [48].

These findings hold particular significance when considering the existing knowledge on the distribution of potentially zoonotic bacterial genera in bats, a field strongly influenced by research biases towards specific geographic regions [49], with a lack of studies in South America. This imbalance in research efforts could lead to a biased understanding of bat microbiomes and viromes, potentially overlooking important variations that exist across different regions [50,51]. It may limit our ability to detect and monitor emerging infectious diseases that could have global implications. To address these consequences, it is important to promote more balanced and comprehensive research efforts across different regions to gain a more holistic understanding of bat populations, their microbiomes, and the potential risks they pose in terms of disease transmission.

The analysis of functional pathways in the gut microbiota of the four bat species revealed significant differences in their metabolic potentials related to their dietary niches. The LEfSe analysis identified a total of 23 metabolic pathways that were differentially abundant among the bat species. These pathways were related to four main metabolic processes, i.e., nucleotide biosynthesis, nucleotide degradation, carbohydrate metabolism, and vitamin B12 metabolism.

The higher abundance of deoxyribonucleotide biosynthesis in hematophagous bats is consistent with their need for rapid DNA synthesis to replenish blood cells, which they obtain from their prey. Adenosine ribonucleotide biosynthesis and UMP biosynthesis were higher in frugivorous and hematophagous bats, which may be related to their diets that require higher metabolic rates. Frugivorous bats require a more extensive digestive process to extract the nutrients from complex carbohydrates, and hematophagous bats require large amounts of energy to support the high protein content of their blood-based diet [52,53]. On the other hand, although the feeding habits of nectarivorous bats require high energy, they primarily eat a diet that is low in fats and proteins but high in simple carbohydrates. This type of diet can be easily converted into energy to support their high metabolic rate per unit of body mass, helping them save energy that would be spent on producing fats and glucose [54]. Nucleotide degradation was higher in hematophagous and insectivorous bats, which may reflect their need to break down nucleotides for energy production.

The higher abundance of predicted vitamin B12 metabolism pathways in the gut microbiota of hematophagous bats in contrast to its lower abundance in frugivorous bats may be related to the respective different abundances of gut bacteria capable of producing vitamin B12 in bats with different diets [55].

Regarding carbohydrate metabolism, the higher abundances of sucrose degradation and pentose phosphate pathways in frugivorous bats are consistent with their reliance on fruits, which are rich in sugars that can be used for energy production [54]. The higher abundances of myo-, chiro-, and scillo-inositol degradations in nectarivorous bats may reflect the importance of inositol as an energy source in this dietary niche. These pathways may help nectarivorous bats break down inositol into smaller molecules that can be used for energy production. The higher abundance of urate biosynthesis in hematophagous and insectivorous bats may be related to their need to eliminate excess nitrogen produced by protein breakdown. The excessively higher abundances of biotin biosynthesis and methanogenesis from acetate in hematophagous bats may be related to their dependence on blood, which is a source of biotin and acetate [56,57].

Overall, the results of the functional pathway analysis suggest that the gut microbiota of the four bat species have different metabolic potentials related to their dietary niches. The differences in the metabolic potentials reflect the adaptation of the gut microbiota to the specific diets of the bat species, which is essential for their survival and health. The results also highlight the importance of studying the functional potential of the gut microbiota in addition to the taxonomic composition to gain a comprehensive understanding of the gut microbiota–host interactions.

Our study’s limitation lies in the relatively small sample size of animals per group, which may constrain the generalizability of our findings. However, increasing the sample size posed challenges, due to the complexities involved in capturing and studying wildlife animals, particularly endangered species like bats. We were restricted to using only 10 animals per group, as per the predetermined experimental design, in compliance with the stringent regulations of Brazilian law governing the capture of wildlife for research purposes. To address this limitation, we conducted a thorough statistical analysis. Additionally, we controlled for significant factors that commonly interfere with wildlife studies and could potentially impact our results, such as species, sex, and physiological condition. Seasonality is another important aspect to consider. Like previous studies comparing bat diets [58], our research spanned multiple seasons, with each bat species collected during a specific season. Factors like food availability, influenced by seasonality and weather, may affect each bat species’ diet differently. While seasonal activity patterns have been observed in various bat studies [59,60], not all bat species exhibit seasonal variations in their diets. For example, a previous study on *S. lilium*, the frugivorous bat examined in our research, did not detect seasonal variations in its diet [58]. Furthermore, our prior study on oxidative stress and antioxidant defense in different organs of the same bat samples suggests that although seasonality may play a role, it likely does not entirely explain the diversity observed in our results. Diet is likely the predominant factor influencing these differences. Therefore, the consistent and meaningful results we obtained offer valuable insights into how diverse diets can impact the gut microbiota diversity and composition of bats.

## 5. Conclusions

In conclusion, this study provides evidence that the feeding habits of bats can influence the diversity and composition of their bacterial communities. The analysis of functional pathways in the gut microbiota of the four bat species revealed significant differences in their metabolic potential related to their dietary niches. In addition, the identification of potentially pathogenic bacteria suggests that the carriage of microbial pathogens by bats may vary depending on feeding habits and host-specific factors. These results have significant implications for the conservation of bat communities, highlighting the need to promote diverse habitats and food sources to support these ecologically important species.

## Figures and Tables

**Figure 1 biology-13-00363-f001:**
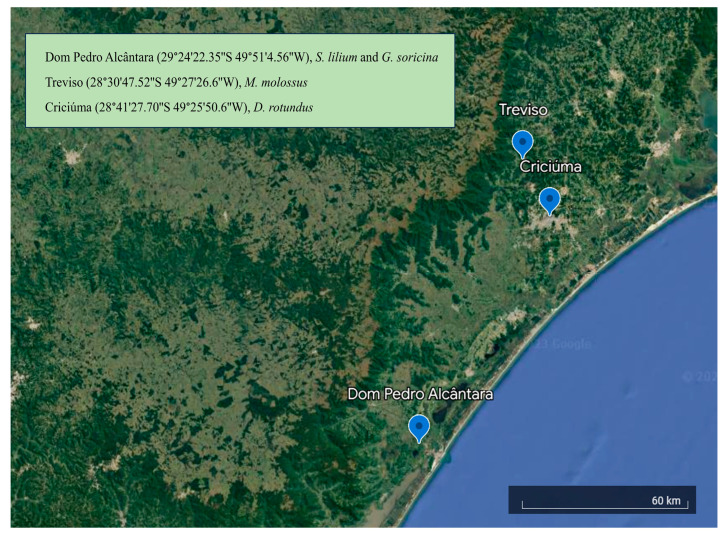
Location of the collection sites and coordinates. Each site is a city in Southern Brazil.

**Figure 2 biology-13-00363-f002:**
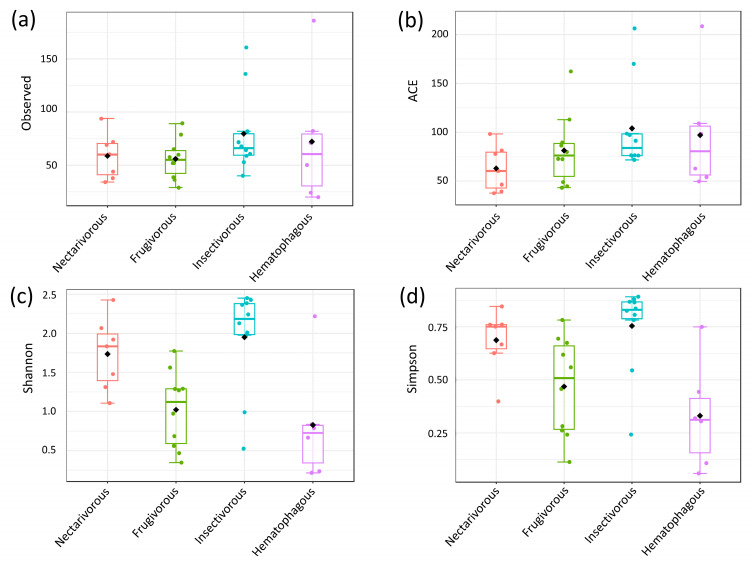
Analysis of alpha diversity based on different metrics. Observed (**a**), ACE (**b**), Shannon (**c**), and Simpson (**d**). Statistical confidence was assessed using the multivariate Kruskal–Wallis test.

**Figure 3 biology-13-00363-f003:**
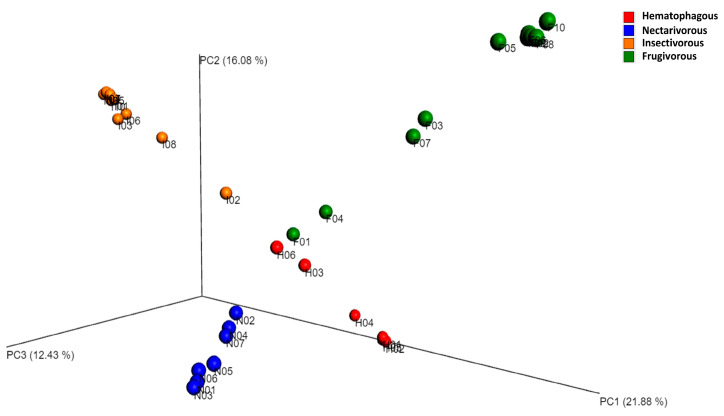
Principal coordinate analysis (PCoA) based on the Bray–Curtis distance metric. The samples were categorized based on the bat species, and to assess the statistical confidence of the sample grouping, ANOSIM was performed.

**Figure 4 biology-13-00363-f004:**
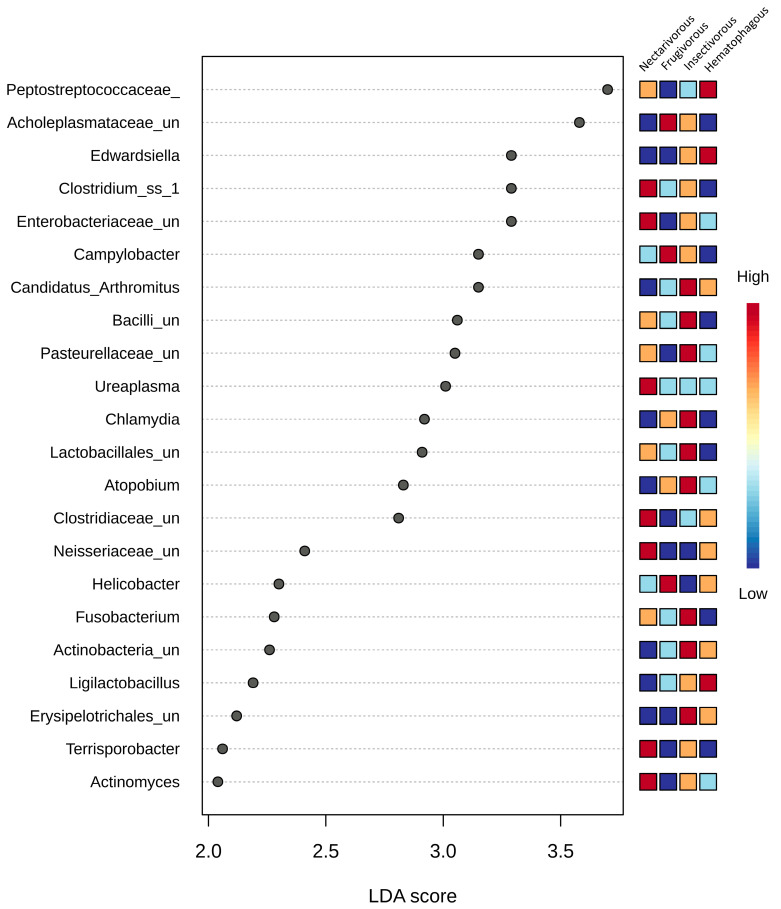
LEfSe analysis conducted on the four bat species at the genus level. Statistical significance was determined using the Kruskal–Wallis test. The microbial taxa were arranged in order based on their LDA score. Differences were considered significant if they had a logarithmic LDA score exceeding ±2.0 and a corrected *p*-value less than 0.1. Colors indicate the correlation changes from negative (blue) to positive (red).

**Figure 5 biology-13-00363-f005:**
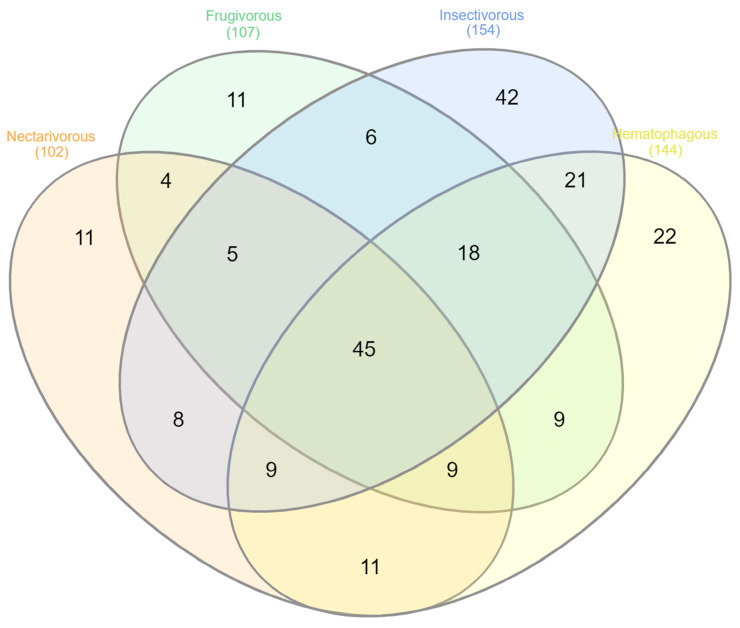
Venn diagram displaying the shared and unique microbial taxa at the genus level across the groups.

**Figure 6 biology-13-00363-f006:**
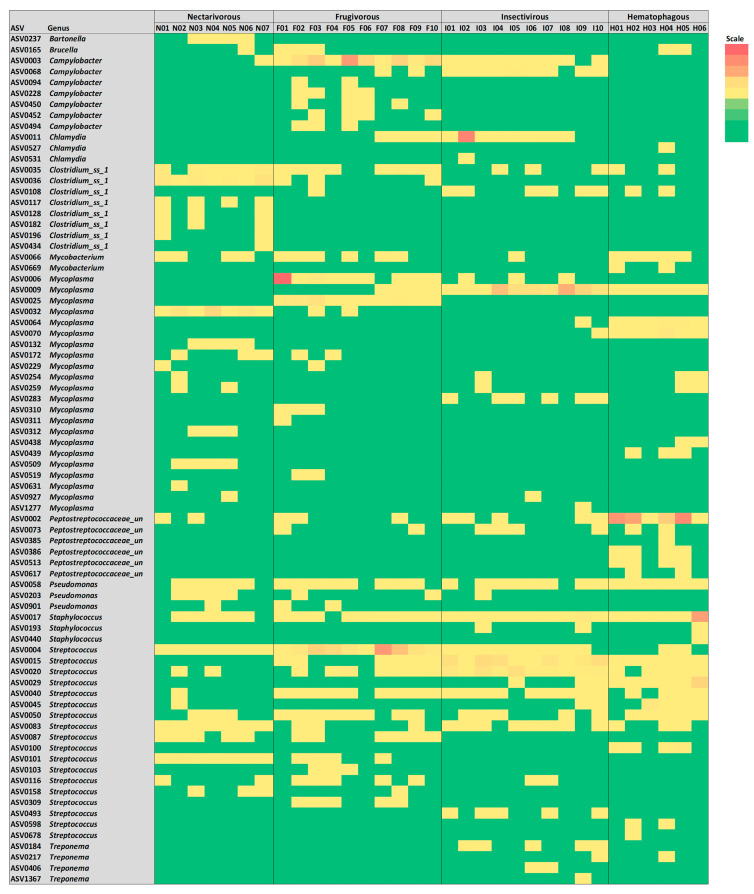
Heatmap of potential bacterial pathogens detected in samples from the four bat species.

**Figure 7 biology-13-00363-f007:**
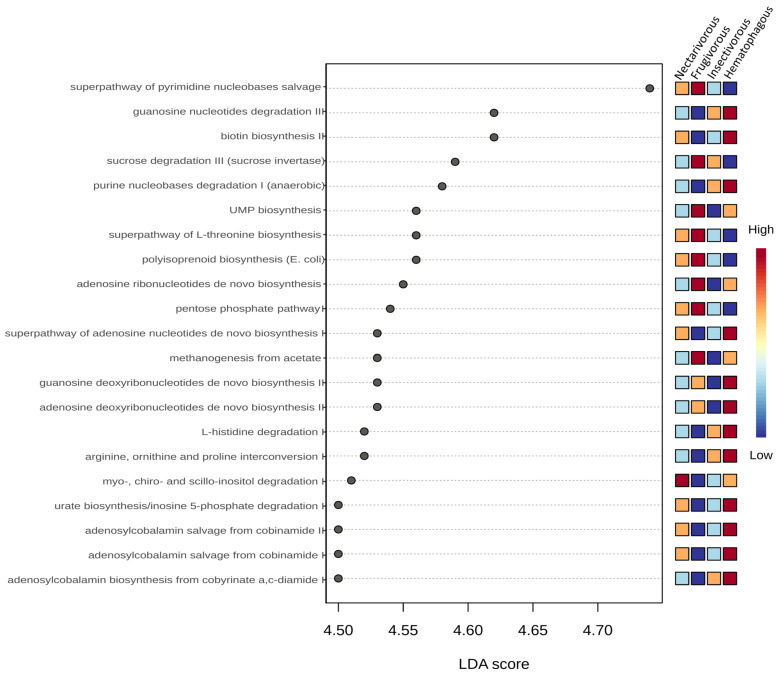
LEfSe analysis of the predicted metabolic pathways. Statistical significance was determined using the Kruskal–Wallis test. The metabolic pathways were arranged in order based on their LDA score. Colors indicate the correlation changes from negative (blue) to positive (red).

## Data Availability

The raw data of high-throughput sequencing is available at the NCBI GenBank Sequence Read Archive (SRA) under accession number PRJNA976745.

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
