# Peer review of "Comparative Analysis of the Gut Microbiota of Bat Species with Different Feeding Habits"

_biology, 2024, doi:10.3390/biology13060363_

Round 1
Reviewer 1 Report (Previous Reviewer 2)
Comments and Suggestions for Authors
The manuscript "Comparative Analysis of the Gut Microbiota of Bat Species with Different Feeding Habits" presents results of an original study of bat gut microbiome analysed with modern bioinformatics methods.
The main question addressed by the research is structural and functional microbiome features in phylogenetically related animals with different types of diet.
All results obtained by the authors are original and relevant for a field of bat gut microbiomes. The paper addresses the specific gap in the field of structure and functional potential of gut microbiome in bats living in South America. The study adds new data on structure, functional potential, similarity and differences between gut microbiomes of bat species with various diets in South America.
The conclusions are in a good agreement with the results and have a scientific soundness. All main questions posed and mentioned above were addressed with modern technique of high-throughput DNA sequencing, namely DNA metabarcoding, following with comprehensive bioinformatic and statistical analysis of the DNA sequences.
The authors have described comprehensively current state of research in bat microbiomes, and have provided all necessary and appropriate references.
The text is well written, the figures are clear and comprehensive. However, resolution of Figures 2, 3, and 6 is insufficient for publishing and should be improved.
Comments on the Quality of English LanguageThe text does not contain serious grammar errors, but I believe that proofreading is desirable.
Author Response
We thank the reviewer for reviewing our manuscript.
Reviewer 2 Report (Previous Reviewer 3)
Comments and Suggestions for Authors
The authors addressed my comments adequately.
However, for the R program, they could use "The R Foundation for Statistical Computing, Vienna, Austria".
Also, the "p" for the "p-value" wasn't italicized throughout the manuscript. For example, please check lines 201-202. I believe they are p-values.
Author Response
We thank the reviewer for the additonal suggestions, which were addressed in the revised version and highlighted in yellow marks.
Reviewer 3 Report (New Reviewer)
Comments and Suggestions for Authors
No comments.
Author Response
We thank the reviewer for reviewing our manuscript.
This manuscript is a resubmission of an earlier submission. The following is a list of the peer review reports and author responses from that submission.
Round 1
Reviewer 1 Report
Comments and Suggestions for Authors
There are minor grammatical mistakes that should be easily fixed with more thorough review.
Reviewer 2 Report
Comments and Suggestions for Authors
The manuscript "Comparative Analysis of the Gut Microbiota of Bat Species with Different Feeding Habits" presents results of an original study, the microbiome data are relevant and analysed with modern bioinformatics methods. The text is well written, the figures are clear and comprehensive. Conclusions are in a good agreement with the results and have a scientific soundness.
However, there are some recommendations below for correction the text. I believe that it will improve the manuscript and make the study more reproducible.
1. Abstract and Introduction would be more sound, if the authors express novelty of their results. It is necessary to demonstrate in clear manner that had been being known by the moment of this study beginning, and what was revealed for the first time by the authors in their study.
2. Lines 111-118. Some details should be given to make the bioinformatic analysis of the raw reads reproducible. Namely, if there were positive (mock community) and negative controls used for NGS of 16S amplicons. Next, the parameters of reads filtering by quality (Q level) and read length should be indicated.
3. Line 156. Is the subtitle name "3.2. Antioxidand Enzymes" a misprint?
4. Line 177 and everywhere. All Latin names of all procaryotic taxa are recommended to mark with Italic font or in another way according to theInternational Code of Nomenclature of Prokaryotes (https://www.microbiologyresearch.org/content/journal/ijsem/10.1099/ijsem.0.005585) See: Chapter 4. Advisory notes A. Suggestions for Authors and Publishers
5. Line 258. The statement "These findings suggest that the different feeding habits may be associated with different ecological niches..." is hardly sounding, because different diets of different species directly indicate their different ecological niches. Moreover, this statement is rather strange taking into account that the bat species studied by the authors have absolutely specific and non-overlapping ecological niches provided by their specific diets and anatomic features of their oral apparatus. There is no suggestion, only obvious fact that the bat species have absolutely different and non-overlapping ecological niches.
6. Lines 357-361. "The observed differences in pathways related to Vitamin B12 metabolism, with higher abundance in hematophagous bats and lower abundance in frugivorous bats, may be related to differences in their gut microbiome. Hematophagous bats have a higher abundance of bacteria that can produce Vitamin B12, whereas frugivorous bats may have less Vitamin B12-producing bacteria due to their different diets [41]." These long sentences can be easely combined and shortened. For example: "The higher abundance of predicted Vitamin B12 metabolism pathway in the gut microbiota of hematophagous bats in contrast to its lower abundance in frugivorous bats may be related to respective different abundances of gut bacteria capable to produce Vitamin B12 in bats with different diets [41]."
7. Lines 380-386. Discussing the limitation of their study the authors did not mention another way of biomaterial collecting without animal killing. Is it possible to capture bats, collect feces and than release the animals? Can such method produce the comparable biomaterial and results compared to the method used by the authors? I believe it would be useful to include this point into Discussion.
Comments on the Quality of English Language
Seems to me that English is good, but I am not a native speaker. Thus, the final decision should be done by the editor.
Reviewer 3 Report
Comments and Suggestions for Authors
In this study, the authors compared the bacterial communities in the gut of four bat species from Southern Brazil to elucidate the relationship between feeding habits, gut microbiota, and bat health. They did a great job. However, I have only one major issue with the bats’ name! I’m requesting the authors to clarify this.
Please find my comments below:
Line 63-66: Please omit them from the introduction section. You could mention them in the discussion and/or conclusion sections.
Line 76-77: As I know, “Glossophaga soricina” should be nectarivore, “Sturnira lilium” should be frugivore, “Molossus molossus” should be insectivore, and the final one is okay. Please correct them. Also, please check all the results based on these four bats!!
Line 115: Please mention the version of VSEARCH here.
Line 119-121: Please provide a reference or database link here.
Line 124: Please provide the company and country name of the R program here.
Line 140,145: “p” from the “p-value” should be in italics. Please correct it throughout the manuscript.
Line 154-155: Where can we get this data? Please provide a supplementary table or something else. If you already have provided it, please mention it here.
Line 160-163: Please transfer this information to the discussion section.
Round 2
Reviewer 3 Report
Comments and Suggestions for Authors
The authors addressed all my comments adequately. I have only one minor comment below:
Line 582-483: Tables S3 and S4 are missing from supplementary materials. Please check this one.
